# Impact of Time-To-Surgery on the Prognosis of Patients with T1 Renal Cell Carcinoma: Implications for the COVID-19 Pandemic

**DOI:** 10.3390/jcm11247517

**Published:** 2022-12-19

**Authors:** Wei Ou, Changxuan Wang, Hiocheng Un, Shengjie Guo, Han Xiao, Bin Huang, Bin Li, Jiahao Lei, Jinhuan Wei, Sui Peng, Junhang Luo, Zongren Wang, Lingwu Chen

**Affiliations:** 1Department of Urology, The First Affiliated Hospital, Sun Yat-sen University, Guangzhou 510080, China; 2Department of Urology, Sun Yat-sen University Cancer Center, Guangzhou 510080, China; 3Clinical Trials Unit, The First Affiliated Hospital, Sun Yat-sen University, Guangzhou 510080, China

**Keywords:** carcinoma, renal cell, nephrectomy, time-to-treatment, survival

## Abstract

Background: During the COVID-19 pandemic, elective surgery has to undergo longer wait times, including nephrectomy for T1 renal cell carcinoma (RCC). This study aimed to investigate the time-to-surgery (TTS) of Chinese T1 RCC patients and its influencing factors, and to illustrate the impact of TTS on the prognosis of T1 RCC. Methods: We retrospectively enrolled 762 Chinese patients with pathological T1 RCC that underwent nephrectomy. To discover the impact of TTS on survival outcomes, we explored the possible delay intervals by week using the Kaplan-Meier method and Log-rank test. Cox proportional hazard models with inverse probability-treatment weighting (IPTW) were used to assess the association between TTS and disease-free survival (DFS) and overall survival (OS). Results: The median TTS of T1 RCC patients was 15 days. The Charlson comorbidity index, the Preoperative Aspects and Dimensions Used for an Anatomical (PADUA) score, and the maximal tumor diameter on presentation were independent influencing factors for TTS. The cut-off point of TTS was selected as 5 weeks according to the Log-rank analysis. For T1a RCC, patients with TTS > 5 weeks had similar DFS (HR = 2.39; 95% CI, 0.82–6.94; *p* = 0.109) and OS (HR = 1.28; 95% CI, 0.23–7.16; *p* = 0.779) compared to patients with TTS ≤ 5 weeks. For T1b RCC, patients with TTS > 5 weeks had shorter DFS (HR = 2.90; 95% CI = 1.46–5.75; *p* = 0.002) and OS (HR = 2.49, 95% CI = 1.09–5.70; *p* = 0.030) than patients with TTS ≤ 5 weeks. Conclusions: Prolonged TTS had no impact on the prognosis of T1a RCC while surgery delayed for over 5 weeks may lead to worse survival in T1b RCC.

## 1. Introduction

The widespread use of radiological techniques has led to an increase in the diagnosis of incidental renal cell carcinoma (RCC), with 70% of cases being in T1 stage [1,2]. The first-line curative treatment for T1 RCC is surgical resection [3]. Patients with T1 RCC in China are eager for prompt surgery, which may impose heavy burdens on limited medical resources. Especially during the COVID-19 pandemic, elective surgery has to undergo longer wait times. But prolonged wait time adds to patient psychosocial stress, leads to gaps in medical care, and may give rise to tumor progression [4]. Therefore, it is crucial to clarify whether the time-to-surgery (TTS) will influence the prognosis of T1 RCC patients.

Prolonged TTS due to patient-, health provider-, and health system-related reasons was reported in many types of malignancies, especially in high-volume national cancer centers [4,5]. Patients frequently voice concerns regarding wait time for cancer treatment [6,7], but its impact on the long-term prognosis varies among cancer types. Previous studies reported that increased TTS led to worse survival in bladder, breast, and colorectal cancer but made no difference to survival in liver or lung cancer [5,8,9]. Whether the TTS is relevant to the prognosis of RCC has not been well demonstrated. So far, four western studies have addressed this issue on disparate stages of RCC but reached different conclusions. Three of them found no significant association between TTS and long-term prognosis of RCC, while longer TTS was associated with worse OS in the fourth study. Nevertheless, they were all limited by recruiting patients from a single center and including long periods of enrollment [10,11,12,13].

Besides, these prior studies also enrolled RCC cases more advanced than T1 stage [10,11,12,13]. Partial nephrectomy is the preferred option for T1 RCC while radical nephrectomy is recommended for T2–T4 RCC [14]. And an increasing number of studies have recommended active surveillance (AS) for T1a RCC due to its indolent biological character [15,16,17,18]. Hence, the TTS and its impact on the prognosis of T1 RCC may be different. Apart from that, the sociocultural background, health resource distribution, hierarchical medical system, and medical insurance system are remarkably different between China and western countries. However, none of the previous studies has focused on T1 RCC in a Chinese setting.

Therefore, the present study was designed to investigate the TTS of T1 RCC patients and its influencing factors with real-world data from a Chinese population, and to illustrate the impact of TTS on prognosis to provide new perspectives for arranging an appropriate treatment schedule in clinical practice.

## 2. Material and Methods

### 2.1. Study Population

The study was performed after the approval of the institutional review board. We retrospectively collected clinical data of 762 consecutive RCC patients from 1 January 2012 to 31 December 2017, including 602 cases from the First Affiliated Hospital of Sun Yat-sen University and 160 cases from the Cancer Center of Sun Yat-sen University. The diagnostic criteria of RCC in the present study were consistent with the European Association of Urology Guidelines [16]. Patients were included if they received initial curative nephrectomy (partial/radical) for T1 RCC confirmed by the postoperative pathological results. Patients were excluded if they received radio frequency ablation (RFA), cryoablation (CA), active surveillance before surgery, or had a history of other malignant tumors, severe organ dysfunction, or mental disorder. To eliminate possible outliers, 53 patients with TTS > 90 days (larger than the 95th percentile of TTS in the whole cohort) were excluded, since their TTS values were discrete. During the follow-up, we failed to collect the clinical information of 95 patients, leaving a final study cohort of 614 patients for survival analysis.

### 2.2. Data Collection and Evaluation

We collected and evaluated the data of 24 clinical variables in the present study.

The values of preoperative lab tests were the latest between date of suspected diagnosis and surgery. The comorbidities were evaluated by the Charlson comorbidity index (CCI) [19]. The anatomical features of RCC were assessed with the Preoperative Aspects and Dimensions Used for an Anatomical (PADUA) scoring system [20] and the postoperative risk stratification of RCC was assessed with the University of California Los Angeles Integrated Staging System (UISS) [21]. The clinical and pathological staging was performed using the 2010 TNM staging system [22]. The surgical complications were evaluated by the Clavien-Dindo classification system [23,24]. The date of suspected diagnosis was defined as the date of the initial detection of tumor lesions (confirmed as RCC thereafter) by ultrasound, computed tomography (CT), or magnetic resonance imaging (MRI). The TTS was defined as the time interval from suspected diagnosis to surgery. The DFS was calculated from nephrectomy to recurrence, metastasis, death, or last follow-up, while the OS was calculated from nephrectomy to all-cause death or last follow-up. Patients were followed at one month after surgery, then semi-annually for the first two years and annually thereafter. The content of the follow-up included clinical manifestations, renal function tests, chest radiography, and abdominal CT.

### 2.3. Statistical Analysis

For ease of interpretation, the continuous variables were categorized according to normal reference values or clinical experience, except for BMI, CCI, and PADUA score. Categorical variables, presented as a number (percentile), were compared using Fisher’s exact test. Continuous variables, reported as medians with the interquartile range (IQR), were compared using Student’s *t* test or Mann-Whitney U test. The univariable and multivariable generalized linear regression models were constructed to identify the influencing factors of TTS. To discover the impact of TTS on DFS and OS, we divided the study cohort into two groups according to the median of TTS. Survival curves of the two groups were generated using the Kaplan-Meier method and compared by the Log-rank test. To analyze the impact of TTS and other observed variables on survival outcomes, univariable and multivariable Cox proportional hazard models were constructed. In the multivariable analysis of TTS and prognosis, other variables with *p* < 0.05 in the univariable analysis were included in the multivariable models. Meanwhile, we utilized the inverse probability of treatment weighting (IPTW) to adjust imbalances of baseline variables between two groups. When analyzing the influence of TTS on short-term prognosis, propensity score matching (PSM) was also used to reduce selection and confounding biases. All baseline variables except for short-term outcomes were matched. The caliper value in PSM was 0.02. All statistical analyses were performed using R-3.6.1 and Stata/MP 14.0 software. All tests were two-sided and *p* < 0.05 indicated statistical difference.

## 3. Results

### 3.1. Patient Characteristics

Patient characteristics are shown in Table 1. The final study cohort included 614 patients with pathological T1 RCC. The median age was 51.6 years old (IQR: 43.3–61.0 years old). The median TTS was 14 days (IQR: 10.0–25.0 days) and 80% had TTS ≤ 4 weeks (Figure 1). The median maximal tumor diameter on presentation was 3.9 cm (IQR: 3.0–5.2 cm) and 45% had maximal tumor diameter on presentation larger than 4 cm. The median follow-up duration of the entire cohort was 43.73 months (IQR: 23.06–64.46 months).

### 3.2. Factors Influencing TTS

Univariable and multivariable generalized linear regression analysis revealed that the Charlson comorbidity index [coefficient = 0.04; 95% confidence interval (CI) = 0.01–0.07; *p* = 0.003], PADUA score (coefficient = −0.02; 95% CI = −0.03–−0.01; *p* = 0.017), and the maximal tumor diameter on presentation (coefficient = −0.10; 95% CI = −0.16–−0.04; *p* = 0.002) were independent influencing factors for TTS (Table 2, Appendix A). Patients who had fewer comorbidities, higher PADUA scores, and larger tumor diameter at first visit waited shorter before surgery.

### 3.3. The Impact of TTS on Long-Term Prognosis

Since the maximal tumor diameter on presentation was an independent influencing factor for TTS, we divided the study cohort into T1a group (maximal tumor diameter on presentation ≤ 4 cm) and T1b group (maximal tumor diameter on presentation > 4 cm). To discover the impact of TTS on long-term prognosis of T1 RCC, we explored the possible delay intervals by week in the range of this cohort. Survival curves of two populations are shown in Figure 2a–d. For T1a RCC, the subsequent Log-rank analysis showed no significant difference in DFS and OS between two populations when the cut-off point of TTS was 1 (DFS: *p* = 0.134; OS: *p* = 0.236), 2 (DFS: *p* = 0.489; OS: *p* = 0.939), 3 (DFS: *p* = 0.194; OS: *p* = 0.550), 4 (DFS: *p* = 0.459; OS: *p* = 0.933), or 5 weeks (DFS: *p* = 0.204; OS: *p* = 0.806). For T1b RCC, the Log-rank analysis also indicated no significant difference in DFS and OS between two populations when the cut-off point of TTS was 1 (DFS: *p* = 0.443; OS: *p* = 0.892), 2 (DFS: *p* = 0.294; OS: *p* = 0.321), 3 (DFS: *p* = 0.203; OS: *p* = 0.617), or 4 weeks (DFS: *p* = 0.051; OS: *p* = 0.230). But we detected a significant difference in DFS and OS between two populations at the cut-off point of 5 weeks (DFS: *p* = 0.002; OS: *p* = 0.039). Therefore, we divided the cohort into the TTS ≤ 5 weeks group and the TTS > 5 weeks group for further investigation.

For T1a RCC, patients in the TTS > 5 weeks group were more likely to have smaller maximal tumor diameter on presentation (*p* < 0.001) and longer operation time (*p* = 0.033) than patients in TTS ≤ 5 weeks group. Other characteristics were similar between the two groups (Appendix A). The univariable and multivariable Cox analysis revealed that the DFS [hazard ratio (HR) = 2.13; 95% CI = 0.70–6.54; *p* = 0.185] and OS (HR = 1.31; 95% CI = 0.27–6.37; *p* = 0.741) were both similar between the TTS ≤ 5 weeks group and the TTS > 5 weeks group (Table 3, Appendix A). Cox analysis with IPTW also showed a non-significant impact of TTS on DFS (>5 weeks vs. ≤5 weeks, HR = 2.39; 95% CI, 0.82–6.94; *p* = 0.109) and OS (>5 weeks vs. ≤5 weeks, HR = 1.28; 95% CI, 0.23–7.16; *p* = 0.779) (Appendix A). The survival analysis also found that surgical complication grade and UISS score were independent predictors for DFS (each *p* < 0.05) in T1a RCC patients (Table 3, Appendix A).

For T1b RCC, all baseline characteristics were similar between the two groups (Appendix A). The univariable and multivariable Cox analysis revealed that patients in the TTS ≤ 5 weeks group had longer DFS (HR = 3.04; 95% CI = 1.43–6.48; *p* = 0.004) and OS (HR = 2.95, 95% CI = 1.15–7.53; *p* = 0.024) than patients in the TTS > 5 weeks group (Table 4, Appendix A). Subsequent Cox analysis with IPTW further validated this conclusion (DFS: HR = 2.90; 95% CI = 1.46–5.75; *p* = 0.002; OS: HR = 2.49, 95% CI = 1.09–5.70; *p* = 0.030) (Appendix A). Apart from the TTS, the survival analysis also found that hemoglobin, platelet, intraoperative bleeding, necrosis in pathological tumor tissue, and UISS score were independent predictors for DFS; CCI, intraoperative bleeding, necrosis in pathological tumor tissue, and UISS score were independent predictors for OS (each *p* < 0.05) in T1b RCC patients (Table 4, Appendix A).

### 3.4. The Impact of TTS on Short-Term Prognosis

After adjusting preoperative characteristics between the two populations by IPTW, the analysis suggested that surgical complication grade (T1a: *p* = 0.380; T1b: *p* = 0.393), surgical wound infection (T1a: *p* = 0.774; T1b: *p* = 0.958), operation time (T1a: *p* = 0.143; T1b: *p* = 0.740), and intraoperative bleeding (T1a: *p* = 0.992; T1b: *p* = 0.150) had no significant difference between the TTS > 5 weeks group and the TTS ≤ 5 weeks group both in T1a and T1b RCC patients (Table 5a,b).

## 4. Discussion

In the current study, the median TTS of Chinese patients with T1 RCC was 15 days. The Charlson comorbidity index (CCI), PADUA score, and the maximal tumor diameter on presentation were independent influencing factors for TTS. The survival analysis revealed that prolonged TTS had no impact on the prognosis of T1a RCC patients while TTS > 5 weeks led to shorter DFS and OS in T1b RCC patients. Besides, the short-term outcomes were similar between the TTS ≤ 5 weeks group and the TTS > 5 weeks group both in T1a and T1b RCC patients.

The median TTS ranged from 20 days to 52 days in RCC patients from western countries [10,11,12,13] However, Chinese patients in our study only underwent a median waiting time of 15 days before surgery. To our knowledge, TTS was mainly determined by doctors and patients, but health system-related factors also played an important role [5]. In western countries, oversight agencies and insurance providers attempt to shift complex cancer operations to high-volume hospitals and designated cancer centers [4]. According to a large-scale American research, the increase in TTS ranged from 4 days to 12 days when patients underwent referral [4]. However, on account of the incomplete hierarchical healthcare system and unevenly distributed medical resources in China, most patients are inclined to visit tertiary hospitals directly [25,26]. Therefore, the TTS in the Chinese setting was remarkably shorter, which may impose heavy burdens on medical resources on tertiary hospitals.

The TTS of malignancies has been an issue of concern for health agencies due to increasing caseload and limited medical resources. Especially during the COVID-19 pandemic, elective surgery has to undergo longer wait times due to lockdowns. Despite no definitive literature to support this, the National Health Service (NHS) in the United Kingdom recommended no more than 9 weeks of waiting time for surgery of RCC in 2015 [27]. For T1a RCC, our study found that prolonged TTS had no impact on prognosis. In accordance with our results, the latest guidelines also recommended active surveillance for elderly and comorbid T1a RCC patients based on a large prospective study that indicated similar OS and cancer-specific survival (CSS) in active surveillance and treatment groups [16,17]. For T1b RCC, our study suggested 5 weeks as a relatively safe time window and found that surgery delayed for over 5 weeks led to shorter DFS and OS. However, four previous studies have also addressed this issue but yielded different conclusions [10,11,12,13]. The primary reasons for the different conclusions lie in the selected study populations and disparate definitions for TTS. To our knowledge, we pioneered the investigation on TTS of patients with T1 RCC in the Chinese setting while the four western studies also enrolled more advanced RCC patients [10,11,12,13]. Apart from that, our conclusion was reinforced by the sufficient clinical variables and the appropriate time span. To reduce selection and confounding bias, we enrolled 24 clinical variables which may influence prognosis. Also of note, imaging technologies, treatment strategies, healthcare policies, and case-volume of hospitals may change over time and lead to variations of TTS [5]. All of the prior studies included long periods of enrollment of over 10 years, while our study only took 6 years [10,11,12,13].

Our study found that patients who had fewer comorbidities, higher PADUA scores, and larger tumor diameter on presentation waited less before surgery. Patients with fewer underlying diseases spend less waiting time for evaluation and treatment of comorbidities. Besides, patients with higher PADUA scores may have larger tumor diameter or suffer the invasion of renal sinus or urinary collecting system, which indicates the adverse tumor characteristics [20]. Similar to the previous studies, the associations between short TTS and larger tumor diameter and higher PADUA scores reflect the surgeon’s increased priority for patients with potential tumor aggressiveness [10,11,12,13]. Among patients undergoing active surveillance, although the majority of RCC cases demonstrate slow growth and low rate of progression, some of them are destined to grow rapidly, become locally invasive, and ultimately metastasize as the waiting time goes on [28]. Therefore, timely surgery is crucial for those RCC patients with potential tumor aggressiveness. However, it remains a challenge for clinicians to accurately identify which tumors may exhibit aggressive phenotypes due to tumor heterogeneity and lack of biomolecular markers [13,28]. Delicate predictive models for tumor growth and progression on early-stage RCC are needed to aid in deciding surgery schedule.

For clinical significance, our study initiated the exploration of TTS in T1 RCC patients based on a large sample from two representative tertiary hospitals in China. We found that prolonged TTS had no impact on the prognosis of T1a RCC patients. This helps to ease patient’s anxiety and relieve stress in tertiary hospitals to some extent. For elderly and comorbid T1a RCC patients, active surveillance is worth considering after thorough examinations and professional assessment. Meanwhile, our study revealed that nephrectomy delay of over 5 weeks undermined the survival of T1b RCC patients. This advocates efficiency promotion and resource integration for hospitals and healthcare agencies to minimize redundant surgery delays. During the global COVID-19 pandemic, many elective urological surgeries have to be delayed due to the limited medical resources and lockdown. Urologists throughout the world have to make difficult choices about which surgery should be delayed and how long it could be delayed. The final decision should be carefully counterbalanced by the medical capacity of hospitals and the impact of delayed treatment on prognosis of patients. Therefore, the present study provides guidance for clinical practitioners to deliver appropriate and timely nephrectomy surgeries during the COVID-19 pandemic.

However, there are still several limitations that warrant consideration. First, consistent with other retrospective analyses, an inherent selection bias was inevitable although IPTW was performed. In addition, there remains lost-to-follow-up bias in our study. We enrolled 762 patients, 102 of whom lost the follow-up information, leaving a study cohort of 660 patients for survival analysis. Hence, more prospective studies are required to validate our conclusions.

## 5. Conclusions

The TTS of T1 RCC patients in China was relatively short. Patients with aggressive tumor characteristics were more likely to receive prompt surgery. The prolonged TTS had no impact on the prognosis of T1a RCC patients while TTS > 5 weeks could have an adverse impact on the survival of T1b RCC patients. This study may provide new perspectives for clinicians to deliver appropriate and timely medical care during the COVID-19 pandemic.

## Figures and Tables

**Figure 1 jcm-11-07517-f001:**
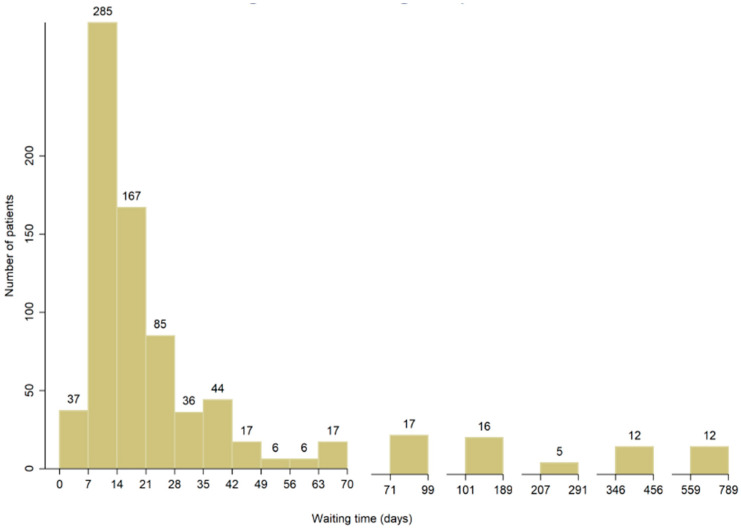
The distribution of TTS among T1 RCC patients.

**Figure 2 jcm-11-07517-f002:**
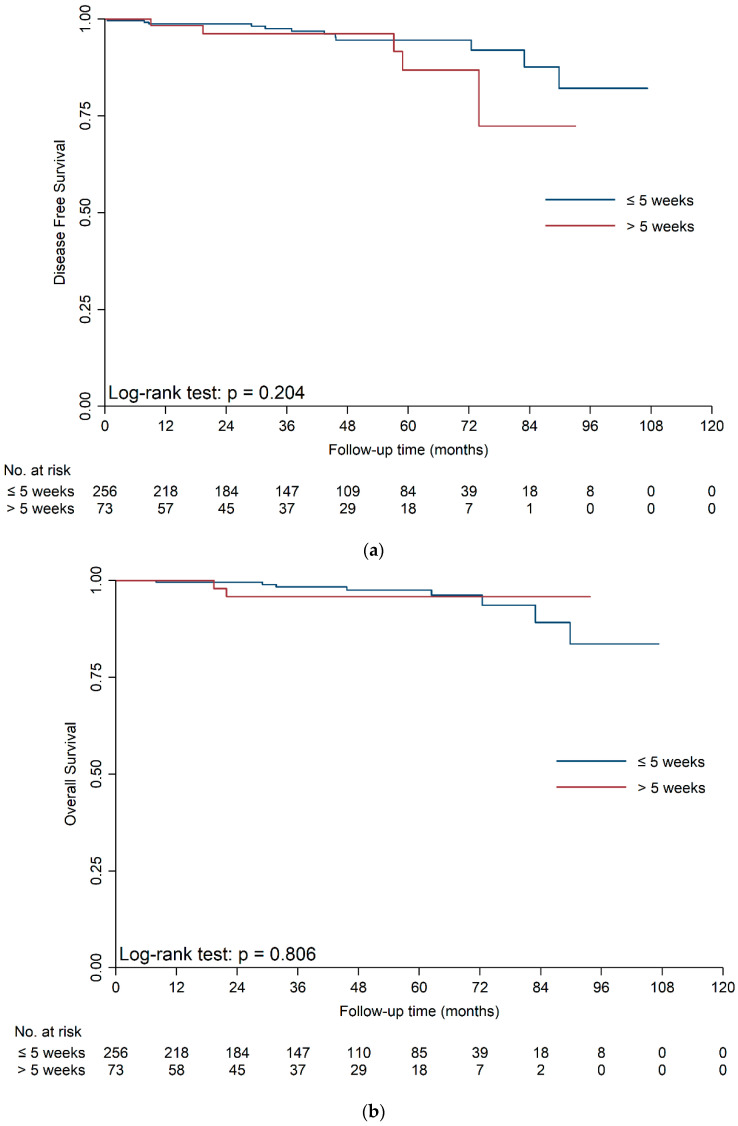
(**a**). The comparison of survival curves between two populations in T1a patients at the cut-off point of TTS = 5 weeks (DFS). Survival curves were generated by Kaplan-Meier method and compared by Log-rank test. (**b**). The comparison of survival curves between two populations in T1a patients at the cut-off point of TTS = 5 weeks (OS). Survival curves were generated by Kaplan-Meier method and compared by Log-rank test. (**c**). The comparison of survival curves between two populations in T1b patients at the cut-off point of TTS = 5 weeks (DFS). Survival curves were generated by Kaplan-Meier method and compared by Log-rank test. (**d**). The comparison of survival curves between two populations in T1b patients at the cut-off point of TTS = 5 weeks (OS). Survival curves were generated by Kaplan-Meier method and compared by Log-rank test.

**Table 1 jcm-11-07517-t001:** Clinical characteristics of all patients.

Characteristics	N = 762
TTS, days, median (IQR)	15.0 (10.0, 27.0)
BMI, kg/m^2^, median (IQR)	23.8 (21.5, 25.5)
CCI, median (IQR)	2.0 (2.0, 2.0)
Age, years, n (%)	
≤60	557 (73.1%)
>60	205 (26.9%)
Sex, n (%)	
Male	502 (65.9%)
Female	260 (34.1%)
ASA score, n (%)	
I	243 (31.9%)
II	456 (59.8%)
III	63 (8.3%)
Maximal tumor diameter on presentation, cm, n (%)	
≤4	428 (56.2%)
>4	334 (43.8%)
Preoperative maximal tumor diameter, cm, n (%)	
≤4	393 (51.6%)
>4	369 (48.4%)
WBC, ×10^9^/L, n (%)	
≤10	715 (93.8%)
>10	47 (6.2%)
Hb, g/L, n (%)	
≤110	52 (6.8%)
>110	710 (93.2%)
PLT, ×10^9^/L, n (%)	
≤100	3 (0.4%)
>100	759 (99.6%)
Serum creatine, μmol/L, n (%)	
≤115	724 (95.0%)
>115	38 (5.0%)
Serum calcium, mmol/L, n (%)	
<2.25	318 (41.7%)
≥2.25	444 (58.3%)
ALP, U/L, n (%)	
≤110	734 (96.3%)
>110	28 (3.7%)
PADUA score, median (IQR)	9.0 (8.0, 10.0)
UISS grade, n (%)	
Low-risk	390 (51.2%)
Intermediate-risk	276 (36.2%)
High-risk	96 (12.6%)
Histopathology, n (%)	
Clear cell RCC	688 (90.3%)
Papillary RCC	54 (7.1%)
Chromophobe RCC	20 (2.6%)
Necrosis in pathological tumor tissue, n (%)	
No	704 (92.4%)
Yes	58 (7.6%)
Surgical approach, n (%)	
Open	235 (30.8%)
Laparoscopic	527 (69.2%)
Surgical resection method, n (%)	
Partial nephrectomy	334 (43.8%)
Radical nephrectomy	428 (56.2%)
Operation time, min, n (%)	
≤150	395 (51.8%)
>150	367 (48.2%)
Intraoperative bleeding, mL, n (%)	
≤50	474 (62.2%)
>50	288 (37.8%)
Surgical complication grade, n (%)	
None	605 (79.4%)
I	134 (17.6%)
II–IV	23 (3.0%)
Surgical wound infection, n (%)	
No	744 (97.6%)
Yes	18 (2.4%)
Postoperative blood transfusion, n (%)	
No	748 (98.2%)
Yes	14 (1.8%)

Statistically significant at alpha = 0.05. Note: Categorical variables are presented as n (%). Continuous variables are described as median (interquartile range). Abbreviations: TTS, time-to-surgery; IQR, interquartile range; BMI, body mass index; CCI, Charlson comorbidity index; ASA, American Society of Anesthesiologists; WBC, white blood cell; Hb, hemoglobin; PLT, platelet; ALP, alkaline phosphatase; PADUA, Preoperative Aspects and Dimensions Used for an Anatomical; UISS, University of California Los Angeles Integrated Staging System.

**Table 2 jcm-11-07517-t002:** Multivariable generalized linear regression of factors influencing TTS among all patients.

Variables	Coefficient	95% CI	*p*-Values
CCI	0.04	(0.01, 0.07)	0.003
PADUA score	−0.02	(−0.03, −0.01)	0.017
Maximal tumor diameter at presentation, cm			
≤4	0		
>4	−0.10	(−0.16, −0.04)	0.002
Preoperative maximal tumor diameter, cm			
≤4	0		
>4	0.05	(−0.01, 0.12)	0.102

Statistically significant at alpha = 0.05. Abbreviations: CI, confidence interval; CCI, Charlson comorbidity index; PADUA, Preoperative Aspects and Dimensions Used for an Anatomical.

**Table 3 jcm-11-07517-t003:** Multivariable Cox regression for DFS and OS among T1a patients.

Variables	DFS			OS		
	HR	95% CI	*p*-Values	HR	95% CI	*p*-Values
TTS group, week						
≤5	1.00			1.00		
>5	2.13	(0.70, 6.54)	0.185	1.31	(0.27, 6.37)	0.741
Age, years						
≤60	1.00			1.00		
>60	1.14	(0.36, 3.62)	0.825	6.39	(0.69, 59.48)	0.103
ASA score						
I	1.00			1.00		
II	1.17	(0.33, 4.15)	0.808	0.87	(0.37, 2.05)	0.747
III	3.86	(0.77, 19.29)	0.100	1.38	(0.50, 3.85)	0.534
Hb, g/L						
≤110	1.00					
>110	0.32	(0.08, 1.24)	0.099			
Surgical complication grade						
None	1.00					
I	5.93	(1.93, 18.25)	0.002			
II–IV	6.05	(0.66, 55.29)	0.111			
UISS grade						
Low-risk	1.00			1.00		
Intermediate-risk	7.44	(1.31, 42.24)	0.024	1.86	(0.10, 35.10)	0.678
High-risk	5.53	(0.81, 37.95)	0.082	1.97	(0.09, 43.98)	0.668

Statistically significant at alpha = 0.05. Abbreviations: DFS, disease-free survival; OS, overall survival; HR, hazard ratio; CI, confidence interval; TTS, time-to-surgery; ASA, American Society of Anesthesiologists; Hb, hemoglobin; UISS, University of California Los Angeles Integrated Staging System.

**Table 4 jcm-11-07517-t004:** Multivariable Cox regression for DFS and OS among T1b patients.

Variables	DFS			OS		
	HR	95% CI	*p*-Values	HR	95% CI	*p*-Values
TTS group						
≤5 weeks	1.00			1.00		
>5 weeks	3.04	(1.43, 6.48)	0.004	2.95	(1.15, 7.53)	0.024
CCI	1.27	(0.97, 1.67)	0.083	1.39	(1.04, 1.86)	0.027
Sex						
Male				1.00		
Female				0.41	(0.16, 1.08)	0.072
Age, years						
≤60	1.00			1.00		
>60	1.88	(0.85, 4.18)	0.120	1.53	(0.63, 3.71)	0.344
ASA score						
I	1.00			1.00		
II	0.88	(0.39, 1.95)	0.745	0.99	(0.36, 2.73)	0.983
III	0.82	(0.27, 2.42)	0.714	1.14	(0.31, 4.16)	0.843
Hb, g/L						
≤110	1.00			1.00		
>110	0.31	(0.14, 0.70)	0.005	0.39	(0.13, 1.19)	0.098
Serum creatine, μmol/L						
≤115				1.00		
>115				0.83	(0.27, 2.56)	0.750
Preoperative maximal tumor diameter, cm						
≤4				1.00		
>4				1.66	(0.58, 4.76)	0.347
Operation time, min						
≤150	1.00			1.00		
>150	1.59	(0.77, 3.30)	0.212	1.41	(0.58, 3.40)	0.450
Intraoperative bleeding, mL						
≤50	1.00			1.00		
>50	2.08	(1.01, 4.29)	0.047	2.49	(1.04, 5.96)	0.041
Surgical complication grade						
None	1.00			1.00		
I	1.64	(0.79, 3.41)	0.182	1.99	(0.86, 4.58)	0.108
II–IV	3.01	(0.79, 11.42)	0.106	4.01	(0.97, 16.49)	0.054
Surgical wound infection						
No	1.00			1.00		
Yes	2.98	(0.75, 11.90)	0.122	3.37	(0.80, 14.15)	0.097
Postoperative blood transfusion						
No	1.00			1.00		
Yes	2.00	(0.17, 23.12)	0.577	4.43	(0.30, 64.54)	0.276
Necrosis in pathological tumor tissue						
No	1.00			1.00		
Yes	3.48	(1.49, 8.16)	0.004	3.61	(1.20, 10.83)	0.022
UISS grade						
Low-risk	1.00			1.00		
Intermediate-risk	1.76	(0.63, 4.92)	0.282	3.24	(0.94, 11.20)	0.063
High-risk	4.72	(1.86, 12.02)	0.001	5.91	(1.82, 19.14)	0.003

Statistically significant at alpha = 0.05. Abbreviations: DFS, disease-free survival; OS, overall survival; HR, hazard ratio; CI, confidence interval; TTS, time-to-surgery; CCI, Charlson comorbidity index; PADUA, Preoperative Aspects and Dimensions Used for an Anatomical; ASA, American Society of Anesthesiologists; Hb, hemoglobin; UISS, University of California Los Angeles Integrated Staging System.

**Table 5 jcm-11-07517-t005:** (**a**). Comparison of short-term outcomes between two groups among T1a patients after IPTW (TTS = 5 weeks). (**b**). Comparison of short-term outcomes between two groups among T1b patients after IPTW (TTS = 5 weeks).

**(a)**	**TTS ≤ 5 Weeks** **(N = 382)**	**TTS > 5 Weeks** **(N = 378.7)**	***p*-Values**	**SMD**
Surgical complication grade, n (%)			0.38	0.186
None	290.3 (76.0)	297.6 (78.6)		
I	76.2 (20.0)	77.0 (20.3)		
II–IV	15.4 (4.0)	4.2 (1.1)		
Surgical wound infection, n (%)			0.774	0.042
No	370.7 (97.0)	364.7 (96.3)		
Yes	11.3 (3.0)	14.0 (3.7)		
Operation time, min, median (IQR)	140.00 (115.00, 175.00)	150.00 (121.79, 185)	0.143	0.137
Intraoperative bleeding, mL, median (IQR)	50.00 (20, 100.00)	50.00 (20.00, 100)	0.992	0.093
**(b)**	**TTS ≤ 5 Weeks** **(N = 378.7)**	**TTS > 5 Weeks** **(N = 387.7)**	***p*-Values**	**SMD**
Surgical complication grade, n (%)			0.393	0.210
None	312.3 (82.5)	311.9 (80.4)		
I	57.5 (15.2)	50.5 (13.0)		
II–IV	8.8 (2.3)	25.3 (6.5)		
Surgical wound infection, n (%)			0.958	0.008
No	371.7 (98.2)	380.2 (98.0)		
Yes	7.0 (1.8)	7.6 (2.0)		
Operation time, min, median (IQR)	157.33 (125.00, 195.00)	150.02 (114.72, 191.41)	0.740	0.121
Intraoperative bleeding, mL, median (IQR)	50.00 (30.00, 100.00)	50.00 (50.00, 180.12)	0.150	0.246

Statistically significant at alpha = 0.05. IPTW was used to control imbalances of the following preoperative variables between the two groups: BMI, CCI, age, sex, ASA score, maximal tumor diameter on presentation, preoperative maximal tumor diameter, WBC, Hb, PLT, serum creatine, serum calcium, ALP, PADUA score, surgical approach, and surgical resection method. Abbreviations: IPTW, inverse probability-treatment weighting; TTS, time-to-surgery; SMD, standardized mean difference; IQR, interquartile range; CCI, Charlson comorbidity index; BMI, body mass index; ASA, American Society of Anesthesiologists; WBC, white blood cell; Hb, hemoglobin; PLT, platelet; ALP, alkaline phosphatase; PADUA, Preoperative Aspects and Dimensions Used for an Anatomical.

## Data Availability

Data can be made available upon reasonable request.

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
