# Peer review of "Impact of Time-To-Surgery on the Prognosis of Patients with T1 Renal Cell Carcinoma: Implications for the COVID-19 Pandemic"

_jcm, 2022, doi:10.3390/jcm11247517_

Round 1
Reviewer 1 Report
This is an interesting study exploring the outcome of 762 consecutive pT1 RCC patients from January 1st 2012 73 to December 31th 2017 and stratified according to TTS.
The number of patients included is adeguate although the two groups are not well balanced.
The study was performed using a population in a pre-COVID era, and this issue should be well specified in the discussion.
The implications of this study in the context of patients treated in the pandemic era should be discussed in more detail, considering that in this particular period a delay in treatment schedule and disease management for other urologic tumors has been observed (PMID: 34771440).
Reviewer 2 Report
Congratulation. It is a very comprehensive and well-written paper on time to surgery for T1 renal cell carcinoma. The only thing I could not find was the histology of the RCC. Authors claim that all important factors influencing the prognosis were included. It was the UISS score, the presence of prognosis but not the histological subtype of the tumor. I very much doubt that they were all clear cell? It is not mentioned in the article and this aspect is very important and should be discussed.
